# Quantum Artificial Neural Network Approach to Derive a Highly Predictive 3D-QSAR Model for Blood–Brain Barrier Passage

**DOI:** 10.3390/ijms222010995

**Published:** 2021-10-12

**Authors:** Taeho Kim, Byoung Hoon You, Songhee Han, Ho Chul Shin, Kee-Choo Chung, Hwangseo Park

**Affiliations:** 1Department of Bioscience and Biotechnology, Sejong University, Kwangjin-gu, Seoul 05006, Korea; tahok@hanmail.net; 2Whan In Pharmaceutical Co., Ltd., 11, Songpa-gu, Seoul 05855, Korea; hoon4131@whanin.com (B.H.Y.); shhan@whanin.com (S.H.); hcshin@whanin.com (H.C.S.)

**Keywords:** blood–brain barrier, 3D-QSAR, structural alignment, molecular ESP descriptor, artificial neural network

## Abstract

A successful passage of the blood–brain barrier (BBB) is an essential prerequisite for the drug molecules designed to act on the central nervous system. The logarithm of blood–brain partitioning (LogBB) has served as an effective index of molecular BBB permeability. Using the three-dimensional (3D) distribution of the molecular electrostatic potential (ESP) as the numerical descriptor, a quantitative structure-activity relationship (QSAR) model termed AlphaQ was derived to predict the molecular LogBB values. To obtain the optimal atomic coordinates of the molecules under investigation, the pairwise 3D structural alignments were conducted in such a way to maximize the quantum mechanical cross correlation between the template and a target molecule. This alignment method has the advantage over the conventional atom-by-atom matching protocol in that the structurally diverse molecules can be analyzed as rigorously as the chemical derivatives with the same scaffold. The inaccuracy problem in the 3D structural alignment was alleviated in a large part by categorizing the molecules into the eight subsets according to the molecular weight. By applying the artificial neural network algorithm to associate the fully quantum mechanical ESP descriptors with the extensive experimental LogBB data, a highly predictive 3D-QSAR model was derived for each molecular subset with a squared correlation coefficient larger than 0.8. Due to the simplicity in model building and the high predictability, AlphaQ is anticipated to serve as an effective computational screening tool for molecular BBB permeability.

## 1. Introduction

It is very restrictive for an external molecule to penetrate the blood–brain barrier (BBB) that separates the central nervous system (CNS) from the systemic circulation. Therefore, the ability of BBB passage is a key characteristic for evaluating the suitability of molecules as a CNS drug candidate [1]. BBB permeability is also an important design factor for non-CNS drugs because their significant exposure to the brain may cause a variety of unwanted side effects including neurological disorders [2].

Molecular BBB permeability is usually quantified with the logarithm of blood–brain partitioning (LogBB), the argument of which refers to the ratio of concentrations in brain and plasma. Various computational modeling methods have been proposed so far to predict the LogBB values of organic molecules. Among them, the development of a quantitative structure-activity relationship (QSAR) model has been most actively pursued due to the straightforwardness of the relation between the molecular descriptors and LogBB [3].

Since the development of the comparative molecular field analysis (CoMFA) method [4], three-dimensional (3D) QSAR approaches have been widely applied in computational drug design. The predictive capability of a 3D-QSAR model is critically dependent on the structural alignments among the molecules as well as on the descriptors required for the numerical representations of individual molecules [5]. Although most molecular descriptors are too imperfect to estimate a variety of pharmacological properties with accuracy, 3D-QSAR models have become more predictive by using the quantum mechanical descriptors rather than those calculated with the empirical potential functions [6,7,8].

A precise alignment of 3D molecular structures is the most important prerequisite for the high performance of 3D-QSAR models. Because even a slight deviation from the perfect molecular overlay may cause large errors in predicting biological activity [9], 3D molecular alignment has remained as the most problematic obstacle that should be overcome in the derivation of an accurate 3D-QSAR model. Hence, a great deal of effort has been devoted to developing an efficient method for aligning the 3D atomic coordinates of structurally diverse molecules. Although most molecular alignment methods were focused on the superposition of similar chemical moieties, some novel techniques have also been proposed to overlap whole molecular structures according to the 3D distribution of physicochemical properties [10,11,12,13,14]. Nonetheless, it has been difficult to overcome the ligand alignment bottleneck in 3D-QSAR models, especially in the case of coping with molecules of varying sizes and shapes.

In recent years, the manifestation of sophisticated computational protocols including machine learning and artificial intelligence (AI) algorithms has shed new light on the development of an efficient QSAR model for drug discovery [15,16]. For example, QSAR models became more accurate by implementing AI algorithms in predicting some pharmacological properties because their functional relations with the numerical molecular descriptors were determined more precisely than in the conventional QSAR approaches [17,18,19,20,21,22]. Machine learning algorithms have also turned out to be efficient in deriving the accurate prediction models for various molecular permeabilities [23,24,25]. AI and machine learning algorithms are thus supposed to replace precedent optimization methods, such as multiple linear regression and partial least square analysis.

In previous work, we established an efficient 3D-QSAR prediction model termed AlphaQ by applying the artificial neural network (ANN) algorithm. As a consequence of combining a rigorous structural alignment method and the quantum mechanical molecular descriptors, the optimized AlphaQ model exhibited a high performance in predicting some biochemical and pharmacological properties [26]. The present study was undertaken to derive an accurate 3D-QSAR model for LogBB prediction through the ANN algorithm to associate the LogBB data with the corresponding quantum mechanical molecular descriptors. A method for alleviating the structural alignment bottleneck in the 3D-QSAR model was also proposed.

## 2. Computational and Experimental Methods

### 2.1. Preparation of the Experimental Dataset to Derive and Validate the LogBB Prediction Model

Molecular LogBB values have often served as a yardstick to assess permeability across the BBB. Therefore, experimental LogBB data for molecules with varying size, shape, and atomic composition were collected from the literature to construct a dataset with which the 3D-QSAR prediction model could be derived and validated appropriately [27,28]. Among these molecules, a total of 406 molecules with molecular weight (MW) ranging from 200 to 600 atomic mass unit (amu) were included in the final dataset.

Although the pattern for structural alignments among the molecules has a critical impact on the performance of the resulting 3D-QSAR prediction model, it is very difficult to obtain an accurate molecular alignment, especially in the case that the dataset involves a broad MW range [29]. This is due in a large part to the difficulty in determining a representative molecule that has to serve as the template to align all the other molecules in a common 3D box. For this reason, the initial molecular dataset was divided into eight subsets to construct 3D-QSAR LogBB prediction models suitable for the molecules within a certain range of molecular weight. The MW range in a given molecular subset was determined in such a way that the molecules with a MW between 250 and 453 amu were equally populated among the six subsets. The smallest molecules with a MW lower than 250 amu and the largest ones with a MW higher than 453 amu were then collected separately to constitute the two additional subsets. The individual molecular subsets included 38–71 elements with MW ranges of 200–250, 251–275, 276–301, 302–323, 324–360, 361–398, 399–453, and 454–600 amu. Some large molecules with a MW higher than 500 amu were also included in the dataset because they have a wide spectrum of LogBB values. As widely adopted in the literature [30,31], 80–85% of the molecules in a subset were selected as the elements of the training set to construct a 3D-QSAR model, while the rest of the molecular elements belonged to the test set to validate the optimized LogBB prediction model. In all eight cases, the fivefold external cross-validation was carried out with five kinds of training and test sets generated at random. The merit of fivefold external cross-validation lies in that selection bias can be avoided by using different training and test sets in all five cases.

### 2.2. Calculations of the 3D Molecular Descriptors

Three-dimensional distribution of the electrostatic potential (ESP) in a molecule containing 2*n* electrons was obtained from its determinantal wavefunction, consisting of *n* molecular orbitals, which were calculated with the ab initio quantum chemical method at the RHF/6-31G** level of theory. Using the individual molecular wavefunctions, the charge density (*ρ*) values were calculated at 3D grid points placed with uniform spacing of 0.212 Å in the common box. The ESP (φ) values were then determined at all the grid points, embedding the molecule by solving the Poisson’s equation.
(1)$${\vec{\nabla }}^{2}\varphi (x,y,z)=\rho (x,y,z)$$

Based on the 3D distribution of ESP in each molecule, the numerical molecular descriptor was constructed preliminarily in the form of a K-dimensional vector comprising the ESP values at the predefined K grid points in the common box. Due to a huge number of grid points, it was necessary to reduce the dimensionality of the initial ESP descriptors to the extent adequate for QSAR modeling. This was performed by principal component analysis (PCA), which has been widely used to project the high-dimensional numerical data in the low-dimensional space by extracting the principal components only [32,33]. The projected ESP vectors were finally defined as the mathematical descriptors of individual molecules, and served as the inputs to derive a 3D-QSAR model for LogBB prediction through the ANN algorithm. Because the reduced molecular ESP descriptors were calculated on a fully quantum mechanical basis, they seemed to outperform the conventional descriptors in terms of correlation with the experimental data.

### 2.3. Pairwise 3D Molecular Alignments

Three-dimensional atomic coordinates for calculating the molecular ESP descriptors were obtained by conducting structural alignments among the molecules in the common rectangular box. The length, width, and height of the common grid box for a molecular subset were determined according to the respective maximum distances along the coordinate axes to encompass the van der Waals volumes of all the molecular elements. To provide a sufficient space for translational and rotational movements during the 3D structural alignments, a marginal distance of 2.7 Å was added to the length, width, and height of the common grid box. Grid points were then placed with uniform spacing of 0.106 Å along the three coordinate axes.

The starting structures for the 3D molecular alignment were prepared from the quantum chemical geometry optimizations at the RHF/6-31G** level of theory. The pairwise structural alignment proceeded by translating and rotating each molecule (target) in a molecular subset so as to maximize the overlap with the representative one (template). The position of the template molecule was fixed in the common grid box. A total of 2000 rotamers along the three axes were taken into account for each target molecule to find the optimal atomic coordinates with respect to the template. These rotamers were prepared by uniformly incremental sampling in the SO(3) rotation group using the Hopf fibration method [34]. To reduce the computational cost, the charge density distribution of a target molecule was calculated exactly for the initial structure only while those of the rotamers were interpolated at all the grid points.

Using 2000 rotamers of a target molecule (*j*) and the corresponding charge density data to reflect the intramolecular electronic redistributions, the optimal structural alignment with the stationary template molecule (*i*) was explored by translating each rotamer. The translational movements were iterated by changing the displacement vectors in such a way as to maximize the quantum mechanical cross correlation (*E_ij_*), which was defined between the ESP of *i* (*φ_i_*(*x, y, z*)) and the charge density of *j* (*ρ**_j_*(*x, y, z*)) as follows.
(2)$$E_{ij}={∭}_{V}\varphi_{i}(x,y,z)\rho_{j}(x,y,z)dV$$

In the physical sense, *E_ij_* corresponds to the energy caused by the repulsive electrostatic interactions between the two molecules. Each *E_ij_* value was calculated in a straightforward way using the fast Fourier transform algorithm [35]. The rotamer with the highest *E_ij_* value was selected as the optimal alignment of *j*, and subsequently used as the input to calculate the molecular ESP descriptor.

### 2.4. Determination of the Template for Multiple Pairwise Alignments

To accomplish the multiple pairwise molecular alignments, it was necessary to determine a representative molecule appropriate for the template with respect to all the other molecules in a subset. The template molecule of each subset was identified with the aid of the distance matrix comprising the distance (*d_ij_*) values for all molecular pairs. More specifically, the *d_ij_* value between the two molecules was given by the difference between the quantum mechanical cross correlation (*E_ij_*) and the average of the self-correlations (*E_ii_* and *E_jj_*) associated with the superposition of two identical molecules.
(3)$$d_{ij}=E_{ij}-\frac{1}{2}\left(E_{ii}+E_{jj}\right)$$

In each molecular subset, the distance matrix was constructed as a yardstick to choose the central molecule that served as a structural template in the pairwise 3D molecular alignments. The determination of the template molecule in a subset began by representing all the molecules with the vertices placed in a network according to the *d_ij_* values. Each vertex in the graph was characterized by a centrality parameter to measure the representativeness of the corresponding molecule [36]. Of the known centrality indices, the betweenness centrality parameter was used in this work because it had been most effective in measuring the influence of a vertex on the information flow [37]. The betweenness centrality parameter of a molecule *i* (*C_i_*) was quantified by the number of the shortest paths involving the vertex *i* out of the total number of the shortest paths for all vertex pairs (*N_paths_*).
(4)$$C_{i}=\frac{\sum_{A}\sum_{B}f_{i}(A,B)}{N_{paths}}$$

Here, *f_i_(A,B)* is 1 and 0 if the vertex *i* is included and excluded in the shortest path between the vertices *A* and *B*, respectively. To identify the node with the highest *C_i_* value in each molecular subset, Monte Carlo simulations were carried out with the random walk algorithm. The central molecules were determined in the same way for all eight molecular subsets, and then served as the structural template in the multiple pairwise molecular alignments.

### 2.5. Derivation of the LogBB Prediction Models with ANN Algorithm

Following the structural alignments and the calculations of molecular ESP descriptors, a 3D-QSAR model for LogBB prediction was derived for each molecular subset using the advanced computational protocols. Among a variety of machine learning and AI tools available for public use, the ANN algorithm was applied in this work in a feed forward fashion with the backpropagation of the error network [38]. The whole network consisted of input, hidden, and output layers as described in Figure 1. The projected ESP vectors of the training-set molecules constituted the neurons in the input layer. All the input neurons (${\hat{I}}_{k}$’s) were combined into a sigmoidal function after multiplying the weighting factors (*w_ki_*’s) to generate the intermediate neurons (${\hat{H}}_{i}$’s) in the hidden layer, which were in turn processed in a similar way to produce a single output neuron ($\hat{O}$). The vector elements of $\hat{O}$ (*O_j_*’s) represented the predicted LogBB values of *N* molecules in the training set.
(5)$${\hat{H}}_{i}=sgm\left(\sum_{k=1}^{N}w_{ki}{\hat{I}}_{k}\right)\;\mathrm{and}\;\hat{O}=sgm\left(\sum_{i=1}^{M}w_{ij}{\hat{H}}_{i}\right)$$

Here, *sgm*(*x*) denotes the sigmoidal function given by (1 + *e^−x^*)^−1^. The output neuron can thus be expressed with the input vectors as follows.
(6)$$\hat{O}=sgm\left(\sum_{i=1}^{M}w_{ij}sgm\left(\sum_{k=1}^{N}w_{ki}{\hat{I}}_{k}\right)\right)$$

For simplicity, the number of neurons in the hidden layer (M) was limited to 3 in the optimization of the 3D-QSAR model for LogBB prediction. All experimental data were transformed to range from 0 to 1 to be processed with the sigmoidal function. These normalized experimental LogBB values functioned as the baseline for optimizing the weighting factors to complete a 3D-QSAR prediction model. The parameterization proceeded with a gradient-based minimization on the error hypersurface (*F*), which was given by the sum of the square differences between the experimental (*D_j_*) and the estimated (*O_j_*) LogBB values of *N* molecules in the training set.
(7)$$F=\sum_{j=1}^{N}{\left(D_{j}-O_{j}\right)}^{2}$$

The *F* value of 10^−4^ was used as the criterion for the convergence of weighting parameters.

### 2.6. Experimental Determination of Molecular LogBB Values Using Mouse Models

The protocols for animal study were approved by the Department of Laboratory Animals, Institutional Animal Care and Use Committee of Whan In Pharm (Approval No. AEC-20080430-0008; Suwon, Korea; Approval date: 30 April 2008). Male ICR mice of 7–8 weeks old weighing 30–32 g were purchased from DooYeol Biotech (Seoul, Korea). All these mice were housed 5 per cage under standard conditions (at 23–25 °C and 55–60% relative humidity) on a 12 h light/dark cycle (light on at 7:00 a.m.) with food and water ad libitum (Envigo 2018S Diets, Envigo). Subjects receiving oral injection were food-deprived overnight.

LogBB measurements began by the oral administration of each drug candidate to five mice (*n* = 5) at a dose of 5 mg/kg. The mice were then euthanized by carbon-dioxide asphyxiation at 5, 30, 60, 120, and 360 min after the oral dose. Blood samples were collected via cardiac puncture using heparinized plastic syringes, and centrifuged for 1 min at 9000 rpm. Subsequently, the whole brain tissue was collected and homogenized (T18 digital ULTRA-TURRAX^®^, IKA, Staufen, Germany) with a phosphate buffer. Two 50 µL aliquots of the supernatant and plasma samples were collected and stored at −70 °C.

Following the sample preparations, 0.1 mL of acetonitrile (Honeywell Burdick & Jackson, Morristown, NJ, USA) containing 20 ng/mL carbamazepine (Sigma, St. Louis, MO, USA) was added to 50 μL of aliquot for each biological sample. Carbamazepine served as an internal standard in the mixture. After vortex mixing and centrifuging at 12000 rpm for 10 min, 10 µL of supernatant was analyzed using LC-MS/MS (API5500, AB Sciex, Foster City, CA, USA). Finally, LogBB values of the ten drug candidate molecules were calculated by the logarithm of a ratio of the area under the time-versus-concentration curve (AUC) in brain and plasma.

## 3. Results and Discussion

With respect to the whole molecular dataset including 406 organic compounds, the MWs and LogBB values ranged from 200 to 600 amu and from −2.69 to 1.64, respectively. Prior to constructing a 3D-QSAR prediction model, molecules with varying shapes and sizes were categorized into eight subgroups according to MW so that the 3D structural alignments could be performed with accuracy. Table 1 lists the characteristics of the eight molecular subsets prepared to derive and validate 3D-QSAR models for LogBB prediction separately. The number of elements was kept almost the same among the subsets except for the two subsets containing the smallest (Subset 1) and the largest (Subset 8) molecules. Each molecular subset was divided further into a training and a test set with a ratio of 4.2:1 on average.

The derivation of a LogBB prediction model termed AlphaQ was initiated by preparing the optimal atomic coordinates of individual molecules in the training and the test set. For this purpose, the pairwise 3D structural alignments were performed in such a way as to maximize the *E_ij_* values with respect to the template molecule. All the molecules were assumed to be neutral in calculating the *E_ij_* values to retain the original electronic and structural features. Figure 2 displays the results of multiple structural alignments among the molecules in the same molecular subset. In all eight cases, the core structures of individual molecules appear to be concentrated in the same region while the sidechains point to different directions. Because *E_ij_* values were calculated on a fully quantum mechanical basis, the AlphaQ program is meritorious over the conventional 3D-QSAR packages in that it can be used to conduct 3D molecular alignments systematically, even in coping with structurally diverse molecules with no identical chemical moiety. Unlike the conventional atom-by-atom matching protocol, however, it is difficult to score the accuracy of 3D structural alignments in the quantitative manner. In the absence of a common chemical group among the molecules, it would be desirable to assess the new structural alignment method by the predictive capability of the resulting 3D-QSAR model.

The reliability of the 3D-QSAR prediction models derived in this work was validated in terms of the correlation between the experimental and the calculated LogBB data. Basically, the squared Pearson correlation coefficient for the training set (R^2^_train_) and that for the test set (R^2^_test_) served as yardsticks to measure the accuracy of the LogBB prediction models. These two statistical parameters can be expressed in the following mathematical forms.
(8)$$R_{train}^{2}=1-\frac{\sum_{i=1}^{train}{\left(y_{i}-{\hat{y}}_{i}\right)}^{2}}{\sum_{i=1}^{train}{\left(y_{i}-{\bar{y}}_{train}\right)}^{2}}\;\mathrm{and}\;R_{test}^{2}=1-\frac{\sum_{i=1}^{test}{\left(y_{i}-{\hat{y}}_{i}\right)}^{2}}{\sum_{i=1}^{test}{\left(y_{i}-{\bar{y}}_{test}\right)}^{2}}$$

Here, $\bar{y}$ is the average of the experimental LogBB data while $y_{i}$ and ${\hat{y}}_{i}$ represent the experimental and calculated LogBB values of molecule *i*, respectively. The summations in the R^2^_train_ and R^2^_test_ parameters run over the molecules in the training and test set, respectively.

Figure 3 shows the linear correlation diagrams for the experimental LogBB values versus those estimated with the AlphaQ prediction model involving the *E_ij_*-based molecular alignments and the quantum mechanical ESP descriptors. For each molecular subset, only the best prediction result generated in the fivefold external cross-validation is presented. The full results are provided in Appendix A along with the molecular elements of the training and the test set in each fold. The AlphaQ models for LogBB prediction appear to be optimized successfully in all eight molecular subsets, as exemplified by the R^2^_train_ values higher than 0.989. This indicates that the ANN parameterization converged well irrespective of the MW range in the training set. In contrast to the close similarity in R^2^_train_ values of the eight molecular subsets, the R^2^_test_ parameters range broadly from 0.820 to 0.963 with the variation of the MW range in the subset. The worst prediction results were obtained for Subset 1 (Figure 3a) and Subset 6 (Figure 3f), which contain compounds with MWs ranging from 200 to 250 and from 361 to 398 amu, respectively. The lowest R^2^_test_ values in the two subsets can be understood in the context that Subset 1 and 6 had the widest range of experimental LogBB values including those lower than −2.0 and those higher than 1.3. In both cases, the predictive capability seems to have increased further by augmenting the dataset with molecules with maximal and minimal LogBB values. Despite some defects in the molecular LogBB dataset, the difference between R^2^_train_ and R^2^_test_ values fell to 0.176 in all eight test cases. This implies that the prevalent overtraining problem was alleviated to a great extent in the present 3D-QSAR models for LogBB prediction.

With respect to the performance of the AlphaQ prediction model for LogBB, it is noteworthy that the R^2^_test_ values for all eight molecular subsets were higher than those obtained with a four-component partial least square analysis using 72 molecular descriptors [27], and with the QSAR prediction model involving the ten molecular descriptors selected by the genetic algorithm [28]. The accuracy enhancement in LogBB prediction by AlphaQ can be attributed in a large part to the adequacy of the structural alignment method using the quantum mechanical *E_ij_* values, because the preparation of optimal molecular atomic coordinates is the first prerequisite for obtaining an accurate 3D-QSAR model. AlphaQ also appeared to produce the better prediction results than conventional 3D-QSAR methods such as CoMFA and comparative molecular similarity index analysis (CoMSIA) in terms of the R^2^_test_ values [39]. This indicates the superiority of the quantum mechanical ESP descriptors to the distribution of steric and electrostatic interaction energies in CoMFA as well as to the molecular property fields in CoMSIA.

To further address the performance of AlphaQ, we also calculated the external predictivity parameter (r^2^_pred_) that has been widely used to quantify the accuracy of statistical prediction methods [40,41]. Mathematically, this parameter is expressed as follows.
(9)$$r_{pred}^{2}=1-\frac{\sum_{i=1}^{test}{\left(y_{i}-{\hat{y}}_{i}\right)}^{2}}{\sum_{i=1}^{test}{\left(y_{i}-{\bar{y}}_{train}\right)}^{2}}$$

Here, $y_{i}$ and ${\hat{y}}_{i}$ are the experimental and calculated data for the test set while ${\bar{y}}_{train}$ denotes the averaged value of the data for the training set. The r^2^_pred_ parameter is meritorious over the corresponding R^2^_test_ value in that the data for the training set can also be reflected in validating a prediction model as well as those for the test set. As shown in Figure 3, the r^2^_pred_ parameters for varying training and test sets ranged from 0.809 to 0.954, which exceeded those yielded in the QSAR prediction models for the biochemical potencies of structurally similar molecules with classical and 3D quantum mechanical descriptors [6,42]. It is also worth noting that the difference between the r^2^_pred_ and R^2^_test_ values was negligible in all eight cases, indicating that the training and test sets would be divided reasonably well in terms of model validations. Thus, both statistical validation parameters support the suitability of the AlphaQ model in estimating the molecular LogBB values.

The relatively high predictive capability of AlphaQ is consistent with the precedent computational finding that the quantum mechanical ESP descriptors would be superior to the 3D molecular interaction fields as well as to the classical 1D descriptors [8,26]. The usefulness of quantum mechanical ESPs as molecular descriptors can be elucidated in the context that the ESP distribution on the molecular surface acts as a determinant for biochemical reactions and intermolecular interactions [43]. For example, the ESP values on the van der Waals surface of a molecule turned out to be highly correlated with the potency of ice recrystallization inhibitors [44]. The 3D ESP descriptors used in this work differ from those of the other groups in that the molecular ESP values were calculated at all 3D grid points in the common box, embedding all the molecules instead of those at the surface points only. This modification seems to be necessary to derive a reliable 3D-QSAR model for predicting complicated biological properties such as LogBB as accurately as simple intermolecular interactions. Such a full 3D distribution of quantum mechanical ESPs is anticipated to be an effective numerical molecular descriptor for estimating a variety of pharmacological properties of drug candidates.

Most probably, the good performance of AlphaQ in all eight test cases for LogBB stems from the categorization of the molecules according to the MW range. This can be understood on the grounds that the size dependence has been a major drawback of 3D molecular structural alignments [45], which has in turn acted as the largest error source that affects the accuracy of a 3D-QSAR model. To address the effect of such an alignment problem on the accuracy of AlphaQ, we also constructed and validated a 3D-QSAR LogBB prediction model using a total of 406 molecules in the whole dataset. Figure 4 shows the linear correlation diagram between the experimental and calculated LogBB values using the training and the test set comprising 328 and 78 molecules, respectively. Although the ANN model for LogBB prediction converged successfully with the associated R^2^_train_ value of 0.997, both R^2^_test_ and r^2^_pred_ values dropped sharply to 0.205 and 0.103, respectively. In this case, the optimized 3D-QSAR model seems to have become elusive as the overtraining problem was too severe. When the correlation diagram in Figure 4 is compared with those for Subsets 1–8 (Figure 3), it can be argued that the poor 3D structural alignments between the molecules with a large difference in MW was responsible for the poor predictive capability of the AlphaQ model derived with all the molecules in the dataset.

Although the LogBB values of some test-set molecules deviated considerably from the corresponding experimental data (Figure 3), it was difficult to further improve the accuracy of AlphaQ either by changing the number of hidden layers in the ANN optimization of the 3D-QSAR model or by upgrading the level of quantum chemical method for calculating the ESP descriptors. The largest errors in LogBB prediction were observed for lorazepam (**1** in Figure 5a) and CID 50599 (**2** in Figure 5b) with absolute unsigned errors of 0.60 and 0.53, respectively. The poor LogBB prediction results for these two molecules exemplify that the accuracy of a QSAR model may be influenced to a great extent even by changing a few molecules in the dataset [27]. If **1** and **2** are excluded from the dataset, for example, the R^2^_test_ values of Subset 1 and Subset 4 increase significantly from 0.820 to 0.877 and from 0.881 to 0.928, respectively. With respect to the poor accuracy in LogBB prediction, it is noteworthy that **1** contains 3-hydroxy-1,3-dihydro-2H-benzo[e][1,4]diazepin-2-one moiety as the molecular core (Figure 5a), which is absent in all the other molecules in the dataset. Therefore, the large error in the estimated LogBB value of **1** would stem from the poor learning of ESP descriptors owing to the rarity of similar functional groups in the training set.

As with other 3D-QSAR prediction models, the limitation of AlphaQ lies in that only a single representative molecular conformation can be taken into account both in 3D structural alignments and in the calculation of ESP descriptors. This restraint is supposed to cause an error in estimating the molecular LogBB values. We note in this regard that **2** can exist in different tautomeric structures, as illustrated in Figure 5b. Only one tautomer possessing an OH moiety on the six-membered aromatic ring was considered in the LogBB prediction on the grounds that its electronic energy calculated at the RHF/6-31G** level of theory is lower than that of the other tautomer by 7.52 kcal/mol. The contribution of the minor tautomer of **2** was thus excluded during the entire course of validating the 3D-QSAR model, which would culminate in a large error in the predicted LogBB value. In general, the exact enumeration of all molecular tautomers is required for the resulting 3D-QSAR model to be precise in predicting the pharmacological properties because the experimental data for model building are prepared under consideration of all the tautomeric states [46]. To enhance the performance of AlphaQ in LogBB prediction, therefore, it seems to be necessary to reflect the contributions of multiple tautomers and conformers both in the 3D structural alignment and in the ESP descriptor calculations.

To estimate the applicability domain of the AlphaQ prediction model for LogBB, the outliers and the high-leverage molecules were determined with the leverage approach [47] using the prediction results. The applicability domain could be visualized explicitly with the two boundaries in the William plot of the standardized residuals of the estimated LogBB values versus the corresponding leverage (*h*) values given by the molecular descriptors. In general, a molecule is considered as an outlier unless the absolute value of its standardized residual is less than three times the standard deviation unit. A molecule also falls outside the applicability domain if the *h* value of the molecule exceeds the warning leverage (*h**) defined as follows.
(10)$$h*=\frac{3p}{n}$$

Here, *p* and *n* are the number of ESP descriptors and the number of molecules in the training set, respectively. Predictions with an *h* value higher than *h** may not be reliable because the results can be regarded as a consequence of extrapolation instead of an exact fit.

The William plots of AlphaQ LogBB prediction models for the eight molecular subsets are displayed in Figure 6. We see that all the molecules in the eight subsets had *h* values substantially lower than *h**. Similarly, the standard residuals of the molecules in Subsets 3 and 4 also appeared to reside between the bordering lines. Judging from the William plots for Subsets 3 and 4, the AlphaQ prediction model for LogBB seems to be reliable, at least for molecules with MWs ranging from 276 and 323, on the grounds that all the associated data points reside in the satisfactory realm. On the other hand, the data points of one or two molecules in Subsets 1, 2, and 4–6 turned out to be outliers as they exhibited standardized residuals beyond the boundaries. Overall, 97.3% of the data points were located in the applicability domain. This indicates that LogBB predictions with AlphaQ may involve a high degree of extrapolation for a small number of molecules.

As an additional validation of the LogBB prediction results, the response permutation test or what is also called Y-scrambling was also carried out to check whether the experimental LogBB values were correlated with the molecular ESP descriptors by chance. The AlphaQ prediction model for LogBB would be regarded as suspect if a high correlation remained between the ESP descriptors and the randomized LogBB values. Table 2 lists the R^2^_train_, R^2^_test_, and r^2^_pred_ parameters obtained after 10% of the experimental LogBB data were permutated at random and regressed with the unchanged molecular descriptors. Although all eight randomized models were optimized well with the high R^2^_train_ values, they became less efficient in LogBB prediction as both R^2^_test_ and r^2^_pred_ parameters decreased significantly when compared to those of the original prediction models (Figure 3). This result confirms that the predictive capability of the AlphaQ model for LogBB stems from a true relationship instead of a correlation by chance.

To further validate the performance of the AlphaQ LogBB prediction model on the experimental basis, we calculated the LogBB values of ten molecules that have been under development as drug candidates in the pharmaceutical industry. Compared in Table 3 are the LogBB values calculated with the optimized AlphaQ model and those obtained by experimental measurements, along with the chemical formula and MWs of the ten molecules under consideration. The corresponding data for **1**, **2**, and carbamazepine are also listed for comparison. It is seen that the calculated LogBB values compare reasonably well with the experimental ones, with a root mean square deviation of 0.297. As shown in Figure 7, the squared linear correlation coefficient (R^2^) between the experimental and calculated LogBB values amounted to 0.843, which is between the R^2^_test_ values for Subset 1 and Subset 6 (Figure 3). These experimental validation results confirm the usefulness of the AlphaQ model in estimating molecular LogBB values.

To the best of our knowledge, AlphaQ may be viewed as the first reliable 3D-QSAR model for predicting the LogBB data of structurally diverse molecules with no common scaffold. In terms of the predictive capability given by the R^2^_test_ value, AlphaQ outperforms quantum mechanical solvation models [48] as well as 2D-QSAR models involving the deep neural network algorithm [49]. Furthermore, AlphaQ has a computational advantage over atomistic statistical simulations and high-level quantum mechanical calculations in the context that a highly predictive model can be derived in a straightforward way using a moderate amount of experimental data. In the presence of such a reliable prediction model for molecular pharmacological properties, the drug discovery process would be facilitated by limiting the candidates in the early stage to the druggable molecules only. An accurate QSAR model has become more valuable due to the global requirement for reducing experiments on animals [50]. The AlphaQ prediction model is therefore anticipated to serve as an effective in silico screening tool for druggable molecules because a vast number of molecules can be evaluated in a high-throughput fashion with the advanced graphics processing unit (GPU) architecture.

Although ab initio quantum chemical calculations were applied both in 3D structural alignments and in ESP descriptor calculations, the AlphaQ model for LogBB prediction has much room for further improvement. Most importantly, a large error may arise because the minor tautomers and conformers of each molecule are excluded in model building, as mentioned above. This problem becomes severe in particular when the dataset contains a number of molecules with many torsional degrees of freedom. Hence, the predictive capability of AlphaQ would be enhanced further upon the implementation of 4D-QSAR formalism to calculate the molecular descriptors under consideration of the conformational diversity in molecules [51]. Since a variety of simulation protocols are available for rigorous conformational sampling, our future research will be focused on the improvement of AlphaQ’s predictability within the 4D-QSAR framework.

## 4. Conclusions

In the attempt to obtain a reliable computational tool for predicting molecular LogBB data, a 3D-QSAR model termed AlphaQ was derived with an ANN algorithm using the distribution of quantum mechanical ESPs as the numerical descriptor for individual molecules. To raise its predictive capability, pairwise 3D structural alignments were carried out in such a way as to maximize the quantum mechanical cross correlation between the template and a target molecule. This alignment method has an advantage over the conventional atom-by-atom matching protocol in that structurally diverse molecules can be analyzed as rigorously as the chemical derivatives of the same scaffold. As in other 3D-QSAR prediction models, the performance of AlphaQ was limited by the difficulty in finding the optimal structural alignment between large and small molecules. This alignment problem was alleviated in a large part by dividing the molecules in the dataset into eight subsets according to MW. As a consequence, a highly predictive QSAR model was derived for each molecular subset, implying the adequacy of the new 3D structural alignment method and the quantum mechanical ESP descriptors in the development of LogBB prediction models. Due to its high predictability and simplicity in model building, AlphaQ is anticipated to serve as an effective computational screening tool for molecular BBB permeability.

## Figures and Tables

**Figure 1 ijms-22-10995-f001:**
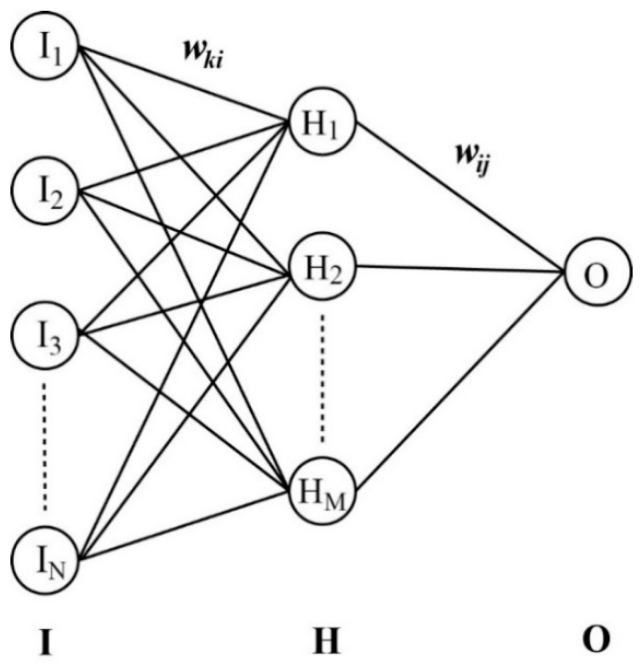
Schematic diagram of N × M × 1 neural network to derive a 3D-QSAR model for LogBB prediction. Columns I, H, and O indicate the input, hidden, and output layers, respectively. Neurons in the three layers are mutually related with the weighting matrices *w_ki_* and *w_ij_*.

**Figure 2 ijms-22-10995-f002:**
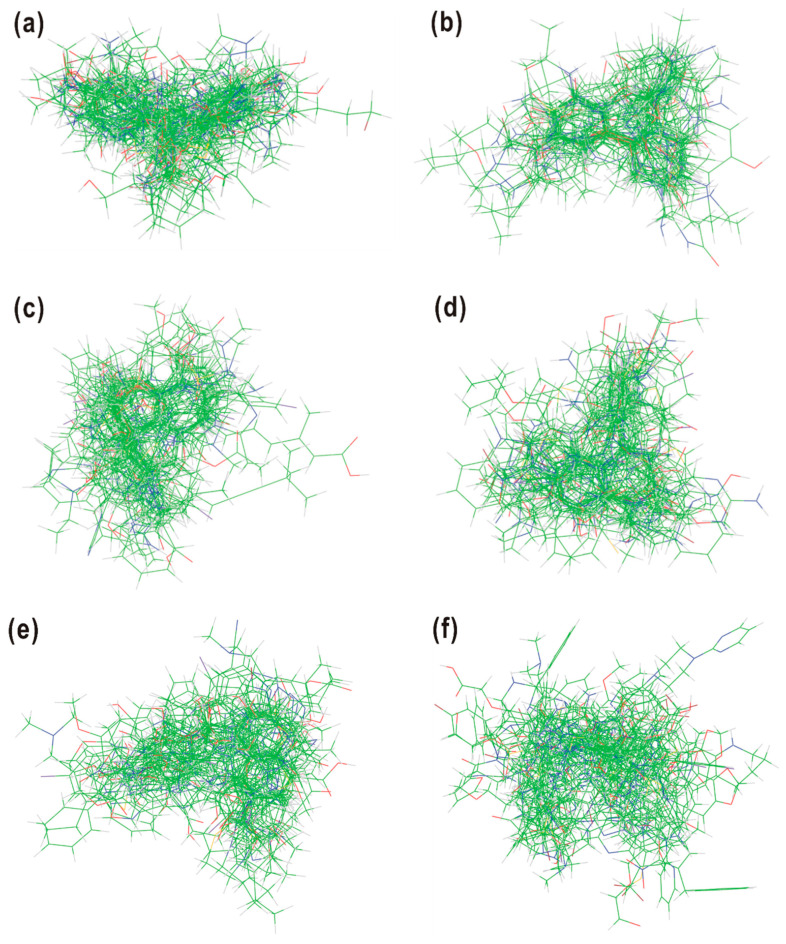
The results of 3D structural alignments among the molecules in (**a**) Subset 1, (**b**) Subset 2, (**c**) Subset 3, (**d**) Subset 4, (**e**) Subset 5, (**f**) Subset 6, (**g**) Subset 7, and (**h**) Subset 8. Carbon, hydrogen, nitrogen, and oxygen are indicated in green, gray, blue, and red, respectively.

**Figure 3 ijms-22-10995-f003:**
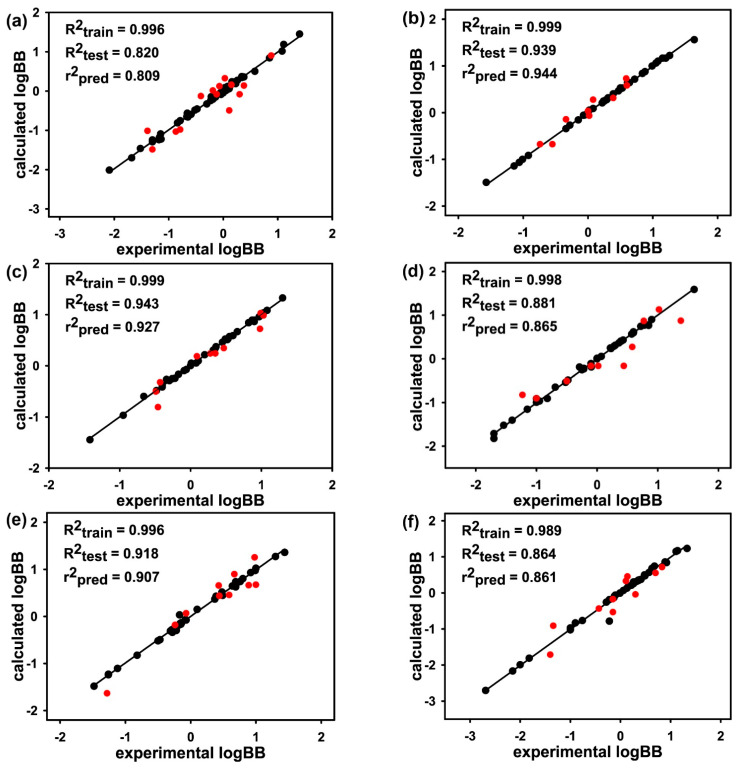
Linear correlation diagram between the experimental and calculated LogBB values for (**a**) Subset 1, (**b**) Subset 2, (**c**) Subset 3, (**d**) Subset 4, (**e**) Subset 5, (**f**) Subset 6, (**g**) Subset 7, and (**h**) Subset 8. Indicated in black and red circles are the molecules in the training and test sets, respectively.

**Figure 4 ijms-22-10995-f004:**
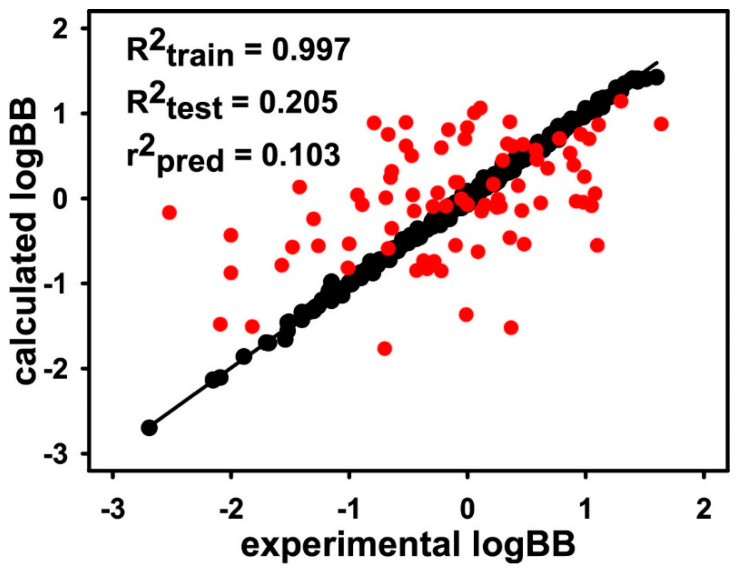
Linear diagram showing correlation between the experimental and calculated LogBB values for a total of 406 molecules in the whole dataset. Indicated in black and red circles are the molecules in the training and test sets, respectively.

**Figure 5 ijms-22-10995-f005:**
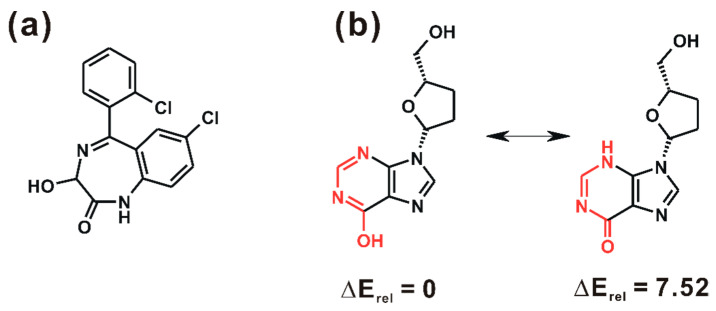
(**a**) Molecular structure of **1** and (**b**) schematic representation of the tautomeric transformation of **2**. Relative electronic energies (ΔE_rel_) calculated at the RHF/6-31G** level of theory are given in kcal/mol. Indicated in red are the atoms involved in the tautomeric change.

**Figure 6 ijms-22-10995-f006:**
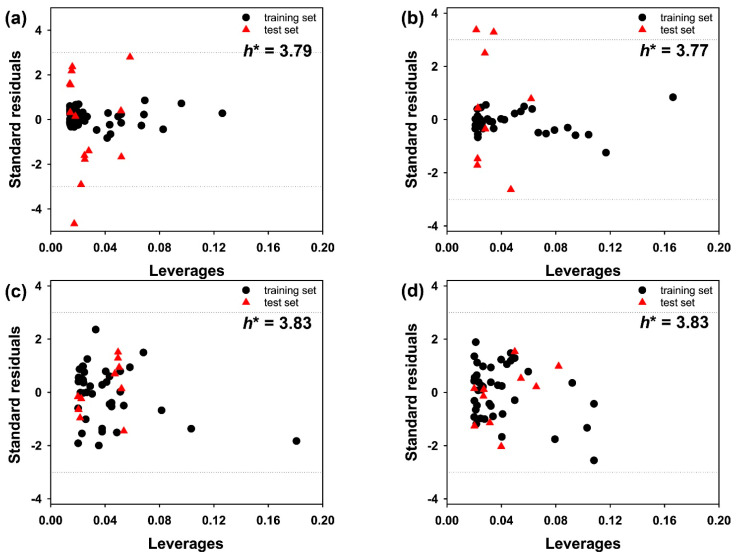
William plots of the LogBB predictions for (**a**) Subset 1, (**b**) Subset 2, (**c**) Subset 3, (**d**) Subset 4, (**e**) Subset 5, (**f**) Subset 6, (**g**) Subset 7, and (**h**) Subset 8. Indicated in black circles and red triangles are the molecules in the training and test sets, respectively. The dashed lines indicate the boundaries of the applicability realm. Warning leverage values (*h**’s) are indicated in numbers instead of vertical lines because they reside too far from the data points.

**Figure 7 ijms-22-10995-f007:**
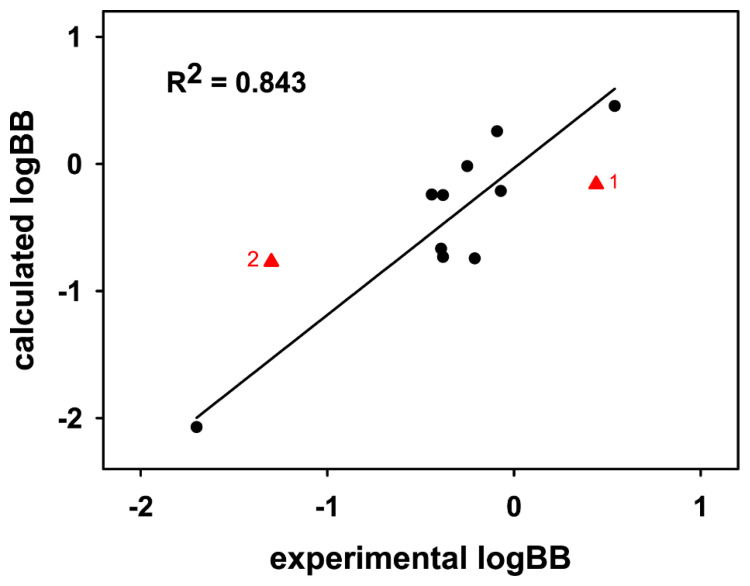
Linear diagram showing correlation between the experimental and calculated LogBB values of the ten drug candidate molecules in comparison with **1** and **2**.

**Table 1 ijms-22-10995-t001:** Characteristics of the eight molecular subsets prepared to derive and validate 3D-QSAR model for LogBB prediction.

Molecular Subset	MW Range	LogBB Range	No. of Training-Set Molecules	No. of Test-Set Molecules
Subset 1	200–250	−2.09–1.40	57	14
Subset 2	251–275	−1.57–1.64	39	9
Subset 3	276–301	−1.42–1.30	40	10
Subset 4	302–323	−1.70–1.60	40	10
Subset 5	324–360	−1.48–1.44	40	10
Subset 6	361–398	−2.69–1.33	40	10
Subset 7	399–453	−1.30–1.51	40	9
Subset 8	454–600	−2.15–1.10	32	6

**Table 2 ijms-22-10995-t002:** Results of the Y-scrambling tests to validate the LogBB prediction models built by shuffling 10% of the experimental LogBB values in the dataset.

	Subset 1	Subset 2	Subset 3	Subset 4	Subset 5	Subset 6	Subset 7	Subset 8
R^2^_train_	0.998	0.997	0.998	0.998	0.997	0.999	0.998	0.997
R^2^_test_	0.592	0.785	0.601	0.748	0.656	0.749	0.692	0.703
r^2^_pred_	0.559	0.694	0.597	0.714	0.655	0.659	0.663	0.658

**Table 3 ijms-22-10995-t003:** Characteristics of the ten molecules used for the experimental validation of the AlphaQ prediction model for LogBB in comparison with those of **1**, **2**, and carbamazepine.

Molecules	Chemical Formula	MW	Calculated LogBB	Experimental LogBB
**1**	C_15_H_10_Cl_2_N_2_O_2_	321.2	−0.16	0.44
**2**	C_10_H_12_N_4_O_3_	236.2	−0.77	−1.30
carbamazepine	C_15_H_12_N_2_O	236.3	−0.04	−0.14
**3**	C_17_H_17_ClN_8_O	384.8	−0.02	−0.25
**4**	C_22_H_25_ClN_6_O_3_	456.9	−0.67	−0.39
**5**	C_23_H_27_ClN_6_O_3_	471.0	−0.24	−0.44
**6**	C_25_H_32_ClN_7_O_2_	498.0	0.26	−0.09
**7**	C_19_H_21_F_4_N_5_O_3_	443.4	−0.21	−0.07
**8**	C_25_H_26_F_3_N_5_O_3_	501.5	−0.74	−0.21
**9**	C_24_H_20_F_3_N_5_O_3_	483.5	−2.07	−1.70
**10**	C_28_H_31_N_7_O_3_	513.6	−0.25	−0.38
**11**	C_17_H_23_N_5_O_2_	329.4	0.46	0.54
**12**	C_24_H_31_N_3_O_2_	393.5	−0.73	−0.38

## Data Availability

The data presented in this study are available in the article or Appendix A.

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
