# Peer review of "Quantum Artificial Neural Network Approach to Derive a Highly Predictive 3D-QSAR Model for Blood–Brain Barrier Passage"

_ijms, 2021, doi:10.3390/ijms222010995_

Round 1

Reviewer 1 Report

In this manuscript, the authors presented a machine learning method to predict the molecular logarithm of the blood-brain partition values. The pairwise 3D structural alignments were used to maximize the quantum mechanical cross-correlation between the template and a target molecule. Then the 3D distribution of the molecular electrostatic potential was used as the numerical descriptor for ANN. Identifying the blood-brain barrier passage of drugs is essential, especially on the central nervous system. The authors used the same method to predict molecular biochemical and pharmacological properties (Physical Chemistry Chemical Physics, 2019). The novelty of this manuscript is (1) applying this method to predict the molecular logarithm of the blood-brain partition values of molecules; (2) building different prediction models based on the molecular weight of compounds. However, there are several concerns that authors need to address.

  1. A precise alignment of 3D molecular structures is the most important for the QSAR model. Hence, it is better to align with the compounds which have a similar size. The authors proposed that 406 molecules were divided into eight subsets according to their molecular weight. It is not clear how to separate these compounds. Why is the interval of the eight subsets different?
  2. The authors used a 3-fold cross-validation method when training the data. The number of molecules in each subset is not large. It is better to use 5-fold or the leave one out method for validation.
  3. Figure 2 displays the results of multiple pairwise structural alignments among the molecules belonging to the same molecular subset. However, the meaning of figure 2 is not clear.
  4. AlphaQ is a program but without any definition or reference.
  5. It is good to use the mouse models to determine the molecular LogBB values experimentally, and the results are shown in Table 2. In this experiment, Carbamazepine was a positive control, but there was no negative control. The results of Carbamazepine should be shown in Table2.
  6. The results of drugs #1 and #2 have to be shown in Table 2 and Figure 6.
  7. The authors should provide the structure of ten molecules(#3~#10) in Table2.

Author Response

1) A precise alignment of 3D molecular structures is the most important for the QSAR model. Hence, it is better to align with the compounds which have a similar size. The authors proposed that 406 molecules were divided into eight subsets according to their molecular weight. It is not clear how to separate these compounds. Why is the interval of the eight subsets different?

With respect to constructing the molecular subsets, the ranges of molecular weight (MW) in the subsets were determined in such a way that the molecules with MW between 250 and 453 amu were equally populated among the six subsets. The smallest molecules with MW lower than 250 amu and the largest ones with MW higher than 453 amu were then collected in separate to constitute the two additional subsets. To explain these, we have added two sentences on p. 5 line 10 from bottom in the revised manuscript.

2) The authors used a 3-fold cross-validation method when training the data. The number of molecules in each subset is not large. It is better to use 5-fold or the leave one out method for validation.

Following the suggestion, we carried out the 5-fold external cross-validation for the better validation of the LogBB prediction models. All the prediction results obtained with five kinds of training and test sets have been presented in Supplementary Material.

3) Figure 2 displays the results of multiple pairwise structural alignments among the molecules belonging to the same molecular subset. However, the meaning of figure 2 is not clear.

Actually, Figure 2 shows the results of 3D molecular alignments obtained with AlphaQ. It is interesting to note that the core structures of individual molecules in each subset are concentrated in the same region while the sidechains point to different directions. This has been clarified on p. 13 line 3 in the revised manuscript.

4) AlphaQ is a program but without any definition or reference.

In accordance with the comment, we have defined the term AlphaQ in the last paragraph on p. 4 along with the reference for the original work (Ref. 26).

5) It is good to use the mouse models to determine the molecular LogBB values experimentally, and the results are shown in Table 2. In this experiment, Carbamazepine was a positive control, but there was no negative control. The results of Carbamazepine should be shown in Table 2.

Following the suggestion, we have presented the experimental and calculated LogBB values of carbamazepine in Table 3 (p. 25) of the revised manuscript.

6) The results of drugs #1 and #2 have to be shown in Table 2 and Figure 6.

As suggested by the reviewer, we have presented the calculated LogBB values of compounds 1 and 2 in Table 3 as well as in Figure 7 in the revised manuscript in comparison with the experimental ones.

7) The authors should provide the structure of ten molecules (#3~#10) in Table 2.

In accordance with the comment, we have presented the structures of compounds 3-12 at the end of Table S1 in Supplementary Material.

Reviewer 2 Report

The manuscript of Park and colleagues concerns the Quantum Artificial Neural Network Approach to Derive a Highly Predictive 3D-QSAR Model for Blood-Brain Barrier Passage. 

The manuscript is interesting and touches on an important feature that a drug should have to act as a CNS drug. 

However, before it could be considered for publication, there are some major issues that the authors have to address.

Major issues:

  • In the introduction section, the authors did not mention the other efforts made in this field to predict the permeability of a compound across different physiological barriers. Just, for example, the review Journal of Chemical Information and Modeling  2019595, 1759-1771, and ACS Omega 2019, 4, 16, 16774–16780, and Chem. Res. Toxicol. 2021, 34, 6, 1456–1467. Please improve the introduction section with these and/or other references.
  • According to the Lipinsky rules of five, it is known that MW>500 uma, could be an issue for the BBB passing. Why did the authors introduce such MW compounds?
  • The materials and methods section should be largely improved, in particular, when the authors describe the methodology of QSAR building model and then its validation. For instance, in line 104, the authors should clarify the meaning of "the 3-fold external cross-validation was carried out with three kinds of training and test sets 105 generated at random.
  • Even though the authors cited the well-known papers of Golbraikh, Tropsha, and Gramatica, the validation metrics they use are quite confused and these are not well described in the material and methods section. The authors cited R2test and R2pred, but R2pred is not defined in the M&M section. Moreover, in what do the differences consist?  
  • Did the authors perform any internal validation such as bootstrapping, Y-scrambling?
  • The authors did not perform any validation about the AD. Please add the Williams plot to demonstrate this.
  • The authors stated the used descriptors outperform the use of classical descriptors. I agree with this statement, but the authors should demonstrate this by introducing the models built with classical descriptors

Author Response

1) In the introduction section, the authors did not mention the other efforts made in this field to predict the permeability of a compound across different physiological barriers. Just, for example, the review Journal of Chemical Information and Modeling 2019, 59, 5, 1759-1771, and ACS Omega 2019, 4, 16, 16774–16780, and Chem. Res. Toxicol. 2021, 34, 6, 1456–1467. Please improve the introduction section with these and/or other references.

Following the suggestion, we have added three references (Refs. 23-25 in the revised manuscript), and placed an emphasis on the fact that the machine learning algorithms were also efficient in deriving the accurate prediction models for various molecular permeabilities on p. 4 line 14.

2) According to the Lipinsky rules of five, it is known that MW>500 uma, could be an issue for the BBB passing. Why did the authors introduce such MW compounds?

With respect to model building and validation, some large molecules with MW higher 500 amu were also included in the dataset because they have a wide spectrum of LogBB values. To explain this, we have added a sentence on p. 5 line 4 from bottom in the revised manuscript.

3) The materials and methods section should be largely improved, in particular, when the authors describe the methodology of QSAR building model and then its validation. For instance, in line 104, the authors should clarify the meaning of "the 3-fold external cross-validation was carried out with three kinds of training and test sets generated at random.

We agreed that further explanation should be provided with respect to the k-fold external cross-validation in QSAR modeling. k was increased to 5 in the revised manuscript for the better validation of the LogBB prediction models. The merit of 5-fold external cross-validation lies in that selection bias can be avoided by using different training and test sets in all five cases. To clarify this, we have added a sentence on p. 6 line 2 in the revised manuscript.

4) Even though the authors cited the well-known papers of Golbraikh, Tropsha, and Gramatica, the validation metrics they use are quite confused and these are not well described in the material and methods section. The authors cited R2test and R2pred, but R2pred is not defined in the M&M section. Moreover, in what do the differences consist?

In accordance with the comment, we have presented the mathematical expression for the r2pred parameter on p. 17 in the revised manuscript. The r2pred parameter is meritorious over the corresponding R2test value in that the data for the training set can also be reflected in validating a prediction model as well as those for the test set. To place an emphasis on this point, we have added a sentence on p. 18 line 5 from bottom.

5) Did the authors perform any internal validation such as bootstrapping, Y-scrambling?

As an additional validation of the LogBB prediction results, the response permutation test or what is also called Y-scrambling was carried out to check whether the experimental LogBB values were correlated with the molecular ESP descriptors by chance. The AlphaQ prediction models for LogBB would be regarded as suspect if a high correlation remains between the ESP descriptors and the randomized LogBB values. To address such a possibility, we obtained the R2train, R2test, and r2pred parameters after 10 percent of the experimental LogBB data were permutated at random and regressed with the unchanged molecular descriptors. Although all eight randomized models were optimized well with the high R2train values, they became less efficient in LogBB prediction as both R2test and r2pred parameters decreased significantly when compared to those of the original prediction models (Figure 3). This result confirmed that the predictive capability of the AlphaQ model for LogBB would stem from a true relationship instead of a correlation by chance. To present and discuss the Y-scrambling results, we have added a paragraph at the end of p. 23 in the revised manuscript along with a table (Table 2).

6) The authors did not perform any validation about the AD. Please add the Williams plot to demonstrate this.

To estimate the applicability domain of the AlphaQ prediction model for LogBB, the outliers and the high-leverage molecules were determined with the leverage approach using the prediction results. The applicability domain could be visualized explicitly with the two boundaries in the William plot of the standardized residuals of the estimated LogBB values versus the corresponding leverage (h) values given by the molecular descriptors. In general, a molecule is considered as an outlier unless the absolute value of its standardized residual is less than 3 times the standard deviation unit. A molecule also falls outside the applicability domain if the h value of the molecule exceeds the warning leverage (h*). The prediction with the h value higher than h* may not be reliable because the results can be regarded as a consequence of the extrapolation instead of the exact fit.

The William plots of AlphaQ LogBB prediction models were obtained for the eight molecular subsets. It was seen that all the molecules in eight subsets had the h values substantially lower than h*. Similarly, the standard residuals of the molecules in Subset 3 and 4 also appeared to reside between the bordering lines. Judging from the William plots for Subset 3 and 4, AlphaQ prediction model for LogBB seemed to be reliable at least for the molecules with MW ranging from 276 and 323 on the grounds that all the associated data points resided in the satisfactory realm. On the other hand, the data points of one or two molecules in Subset 1, 2, and 4-6 turned out to be the outliers as they exhibited the standardized residuals above the boundaries. Overall, 97.3% of the whole data points were located in the applicability domain. This indicated that LogBB predictions with AlphaQ might involve the high degree of extrapolation only for a few number of molecules. The newly obtained results of William plots have been presented and discussed extensively on pp 21-23 in the revised manuscript along with a figure (Figure 6).

7) The authors stated the used descriptors outperform the use of classical descriptors. I agree with this statement, but the authors should demonstrate this by introducing the models built with classical descriptors.

With respect to the performance of the new molecular descriptors, we noted that the AlphaQ program produced the better prediction results than the conventional 3D-QSAR methods such as CoMFA and CoMSIA in terms of the R2test values. This indicates the superiority of the quantum mechanical ESP descriptors to the distribution of steric and electrostatic interaction energies in CoMFA as well as to the molecular property fields in CoMSIA. To explain these, we have added two sentences on p. 17 line 6 along with an additional reference (Ref. 39) in the revised manuscript.

Round 2

Reviewer 1 Report

The authors clearly address my concerns, no further comments are required.

Author Response

Thank you for the good comments.

Reviewer 2 Report

The authors have improved the manuscript and they replied to almost the issues I have raised. I have just a concern before to endorse the publication of this manuscript.

Please add the formulas of R2train, R2test to clarify better the differences in these metrics 

Author Response

Following the suggestion, the mathematical expressions of the R2train and R2test parameters have presented at the end of p. 14 in the revised manuscript.